# Efficacy and safety of Xuebijing injection for radiation pneumonitis: A meta-analysis

**Zheng Li**[1◦], **Dandan Wang**[2◦], **Ying Zhang**[1◦], **Shuo Wang**[3], **Xueqian Wang**[1], **Yuxiao Li**[2], **Yuerong Gui**[2], **Jun Dong**[2], **Wei Hou**[1]*

**1** Department of Oncology, Guang'an men Hospital, China Academy of Chinese Medical Sciences, Beijing, China, **2** Graduate School of Chinese Medicine, China Academy of Chinese Medical Sciences, Beijing, China, **3** Graduate School of Chinese Medicine, Beijing University of Chinese Medicine, Beijing, China

◦ These authors contributed equally to this work.
* houwei1964@163.com

## Abstract

### Background

Currently, the treatment of radiation pneumonitis (RP) remains a clinical challenge. Although glucocorticoids are used for RP treatment, they have associated side effects. Xuebijing injection (XBJ) has been widely used for RP treatment in China, but so far no meta-analysis has evaluated its efficacy and safety.

### Methods

PubMed, Cochrane Library, EMBASE, China National Knowledge Infrastructure, WAN-FANG database, SinoMED, and China Science and Technology J Database were searched for randomized controlled trials related to XBJ in RP treatment. Two researchers independently conducted literature screening, data extraction, and risk of bias assessment. The outcomes were synthesized and analyzed using the Cochrane Review Manager (RevMan 5.3) software, and a forest plot generated.

### Result

Eight articles met the eligibility criteria for further data extraction and meta-analysis. A total of 578 patients with RP participated in these studies, including 296 in the experimental group (XBJ+BT), and 282 in the control group (BT). The results of the meta-analysis revealed that compared to the BT group, XBJ+BT significantly increased the total effective rate (n = 578; RR = 1.45, 95% CI: 1.30 to 1.61, p<0.0001), and IL-10 expression (n = 296; MD = 17.62, 95% CI:13.95 to 21.29, p<0.00001), decreased interleukin-6 (IL-6) expression (n = 296; MD = -21.56, 95% CI:-27.37 to -15.76, p<0.00001), that of tumor necrosis factor alpha (n = 246; MD = -25.63, 95% CI:-30.77 to -20.50, p<0.00001), and that of C-reactive protein (n = 296; MD = -48.61, 95% CI:-56.49–40.73, p< 0.00001).

### Conclusion

Based on our results, we do not recommend XBJ as an adjuvant treatment for RP. Further randomized controlled trials with rigorous design, strict implementation, and standard reporting are needed to further evaluate the efficacy and safety of XBJ for RP treatment.

**Data Availability Statement:** All relevant data are within the paper and its Supporting Information files.

**Funding:** This project is funded by the Priority theme setting and evidence-based research

scheme design of Traditional Chinese Medicine for malignant tumors (No. K858), National Natural Science Foundation of China (No. 82004179), and the Beijing Municipal Natural Science Foundation (No. 7192181). The funders had no role in study design, data collection and analysis, decision to publish, or preparation of the manuscript.

**Competing interests:** The authors have declared that no competing interests exist.

**Abbreviations:** RT, radiation therapy; RILI, radiation-induced lung injury; RP, radiation pneumonitis; RF, radiation-induced fibrosis; TCM, traditional Chinese medicine; CAM, complementary and alternative medicine; XBJ, Xuebijing injection; PRISMA, Preferred Reported Items for Systematic Review and Meta-analysis; RCTs, randomized controlled trials; CNKI, China National Knowledge Infrastructure; IL-6, interleukin-6; IL-10, interleukin-10; TNF-α, tumor necrosis factor alpha; CRP, C-reactive protein; BT, the control group; XBJ +BT, the experimental group; AEs, adverse events; MD, mean difference; CI, confidence interval; RR, risk ratio.

## Systematic review registration

INPLASY registration number: INPLASY2020120037.

## Introduction

Radiation therapy (RT) is traditionally regarded as a method for local tumor treatment [1]. It is estimated that 10–30% of patients with thoracic malignant tumors receiving radiation therapy develop radiation-induced lung injury (RILI) [2], which consists of two stages: radiation pneumonitis (RP) and radiation pulmonary fibrosis (RF). Once RF is reached, the condition is almost irreversible; therefore, the treatment of RP is particularly important. Patients with RILI may experience progressive dyspnea, worsening lung function, and respiratory failure, eventually leading to death [3]. Currently, clinical treatment is not required for asymptomatic RP. The treatment of symptomatic RP with CTCAE 4.0 grade $\geq 2$ includes glucocorticoids, antibiotics, oxygen, symptomatic treatment, and mechanical ventilation support, if necessary [4]. However, a large dose of glucocorticoids can easily induce or aggravate infections, leading to osteoporosis, digestive and cardiovascular complications, and increased intraocular pressure [5, 6], which affect the patients' quality of life.

In China, traditional Chinese medicine (TCM) treatment is under the guidance of unique Chinese medical theories, which has a history of thousands of years to prevent and treat diseases. TCM treatment, as one of complementary and alternative medicine (CAM), includes Chinese medicine decoction, Chinese patent medicine, traditional Chinese medicine (TCM) injection, acupuncture, Tai Chi and so on. In the treatment of RP, TCM treatment can also improve symptoms such as cough, wheezing, and fever, which has gradually attracted the researchers' attention [7]. Xuebijing injection (XBJ) is one of TCM injections commonly used in clinical prescriptions in China. Its main components are *Carthami flos*, *Radix paeoniae rubra*, *Angelicae sinensis radix*, and *Radix salviae chuanxiong* [8]. These herbs can activate blood circulation, and resolve blood stasis, and detoxification, and are used to relieve fever, heart palpitations, wheezing, other stasis symptoms, and the toxic mutual syndrome caused by systemic inflammatory response syndrome [9, 10].

A number of randomized controlled trials (RCTs) have indicated that XBJ combined with basic treatment (BT) could be advantageous for RP treatment [11, 12]. However, there is a lack of reliable evidence-based medicine, and questions about its effectiveness and safety have not been accurately answered. Therefore, the main purpose of our meta-analysis was to: (1) evaluate the efficacy of XBJ in the treatment of RP compared with basic medical treatment methods and (2) evaluate the safety of the clinical use of XBJ.

## Methods

### Protocol registration

The study protocol was registered on INPLASY (ID:INPLASY2020120037, https://inplasy.com/inplasy-2020-12-0037/)) and previously published [13]. This meta-analysis was implemented based on the Preferred Reporting Items for Systematic Review and Meta-analyses Statement (PRISMA) [14].

### Eligibility criteria

**Type of studies.** Only RCTs associated with XBJ for RP treatment were included. Other studies, such as animal experiments, case reports, reviews, cohort studies, case-control studies, and commentaries were excluded.

**Type of participants.** Participants >18 years of age who were clinically, pathologically, or histologically diagnosed with RP were included. Participants with chronic bronchitis, chronic obstructive pulmonary disease, RF, acute infectious disease, severe heart disease, and KPS<60 were excluded. Gender, nationality, and ethnicity were not restricted.

**Type of interventions.** The interventions were divided into two groups: the control group (BT) and the experimental group (BT and XBJ, XBJ+BT). The former contained glucocorticoids along with bronchodilators, antibiotics, oxygen therapy, and respiratory support therapy when necessary. The latter included XBJ and the interventions in the control group. There were no restrictions on dosage, intervention time, or pharmaceutical provider.

**Type of outcome.** The primary outcome was the total effective rate, which equaled the ratio of patients cured and improved to the total number of patients [15]. The clinical efficacy evaluation includes cure, improvement and failure. Cure: all clinical symptoms disappeared and lung CT examination showed that pulmonary inflammation was completely absorbed; improvement: most of clinical symptoms disappeared and lung CT examination showed that pulmonary inflammation was partially absorbed but there were some fibrous shadows; failure: clinical symptoms did not disappear and lung CT examination showed that the pulmonary inflammation was not absorbed and extensive fibrous shadow distribution was seen and even respiratory failure occurred.

Secondary outcomes were inflammatory cytokines, including interleukin-6 (IL-6), interleukin-10 (IL-10), tumor necrosis factor-alpha (TNF-$\alpha$), C-reactive protein (CRP), and the incidence of adverse events (AEs).

## Search strategy

The following seven electronic databases were searched: PubMed, Cochrane Library, Embase, China National Knowledge Infrastructure (CNKI), WANFANG database, SinoMED, and China Science and Technology J Database. The search time included from the beginning of these libraries to May 1, 2021. There were no language limitations. In addition, two clinical trial registry websites (http://www.chictr.org and http://www.clinicaltrials.gov) were also searched for unpublished clinical trials. The search keywords were: radiation pneumonitis, Xuebijing injection, and randomized controlled trial; for more details, see Table 1.

## Study selection

Two reviewers independently conducted the study selection process. Study titles were screened and duplicates eliminated using NOTEEXPRESS software (version 3.0). Then, abstracts and articles were carefully read, and studies not conforming to the eligibility criteria were removed. In case of disagreement during the process, a third reviewer helped with the evaluation.

**Table 1. Search strategy for PubMed.**

| Number | Search terms |
| --- | --- |
| #1 | Radiation Pneumonitis[MESH] |
| #2 | (Radiation Pneumonitides) OR (Pneumonitides, Radiation) OR (Pneumonia, Radiation) OR (Pneumonias, Radiation) OR (Radiation Pneumonias) OR (Pneumonitis, Radiation) OR (Radiation Pneumonia) OR (Fibrosis, Radiation) OR (Radiation Fibrosis) OR (Radiation induced lung injury) OR (Radiation induced pneumonitis) |
| #3 | (XUEBIJING) OR (XUE-BI-JING) OR (XUEBIJING injection) OR (XUE-BI-JING injection) |
| #4 | (Randomized Controlled Trials[Mesh])OR(Clinical Trials, Randomized) OR (Trials, Randomized Clinical) OR (Controlled Clinical Trials, Randomized) OR (RCT) |
| #5 | #1 AND #2 AND #3 AND #4 |

### Data extraction

Data were independently extracted from the preliminary articles screened by two reviewers. Any disagreement should be resolved in consultation with a third reviewer. The extracted data included the first author's name, publication year, country, sample size, patient characteristics, intervention measures, primary and secondary outcomes, and adverse events. Missing data were obtained, if possible, by contacting the original author.

### Risk of bias assessment

The "risk of bias assessment tool" of the Cochrane Handbook was applied to evaluate the risk of bias of the included articles [16]. Any objections were resolved through discussion or negotiation with a third reviewer. The assessment criteria were as follows: random sequence generation (selection bias), allocation concealment (selection bias), blinding of participants and personnel (performance bias), blinding of outcome assessment (detection bias), incomplete outcome data (attrition bias), selective reporting (reporting bias), and other biases.

### Data analysis

Review Manager software (version 5.3) was used for data processing. The total effective rate was expressed as the risk ratio (RR) with a 95% confidence interval (95% CI). Inflammatory cytokines are shown as mean difference (MD) with 95% CI. The $I^2$ value was applied to assess heterogeneity, and a fixed-effects model used if $I^2 < 50\%$. If $I^2 > 50\%$, the random-effects model was employed, and the source of heterogeneity was explored by subgroup and sensitivity analyses. If >10 articles were included, a stratified analysis was conducted according to intervention measures and participants' characteristics. In the sensitivity analysis, every study was deleted one-by-one to observe the impact on the overall results.

### Publication bias

If there were >10 articles for analysis, a funnel plot was generated to evaluate publication bias [17].

## Results

### Study selection

According to the search strategy, a total of 50 relevant articles were initially screened in seven electronic databases. After removing 35 duplicates using NOTEEXPRESS software, 15 articles remained. By carefully reading the abstracts and full texts, seven articles were removed for the following reasons: animal study (n = 1), wrong study design (n = 4), wrong participants (n = 1), and wrong outcomes (n = 1). Finally, eight articles [18–25] met the eligibility criteria for further data extraction and meta-analysis. Fig 1 illustrates the study selection process.

### Study characteristics

Table 2 shows the characteristics of the eight RCTs included in this study. All of them were conducted in China, and all patients were Chinese. A total of 578 patients with RP participated in these studies, including 296 in the XBJ+BT group and 282 in the BT group. There were 334 male and 224 female patients with ages ranging from 45 to 60 years. Sample sizes varied from 48 to 140. In the BT group, the main treatment was hormonal, and antibiotics were administered in case of infection. Nutritional support, bronchodilators, and oxygen support were provided when necessary. Prednisone was reported in six articles [18–21, 23, 25], five [18–21, 23]

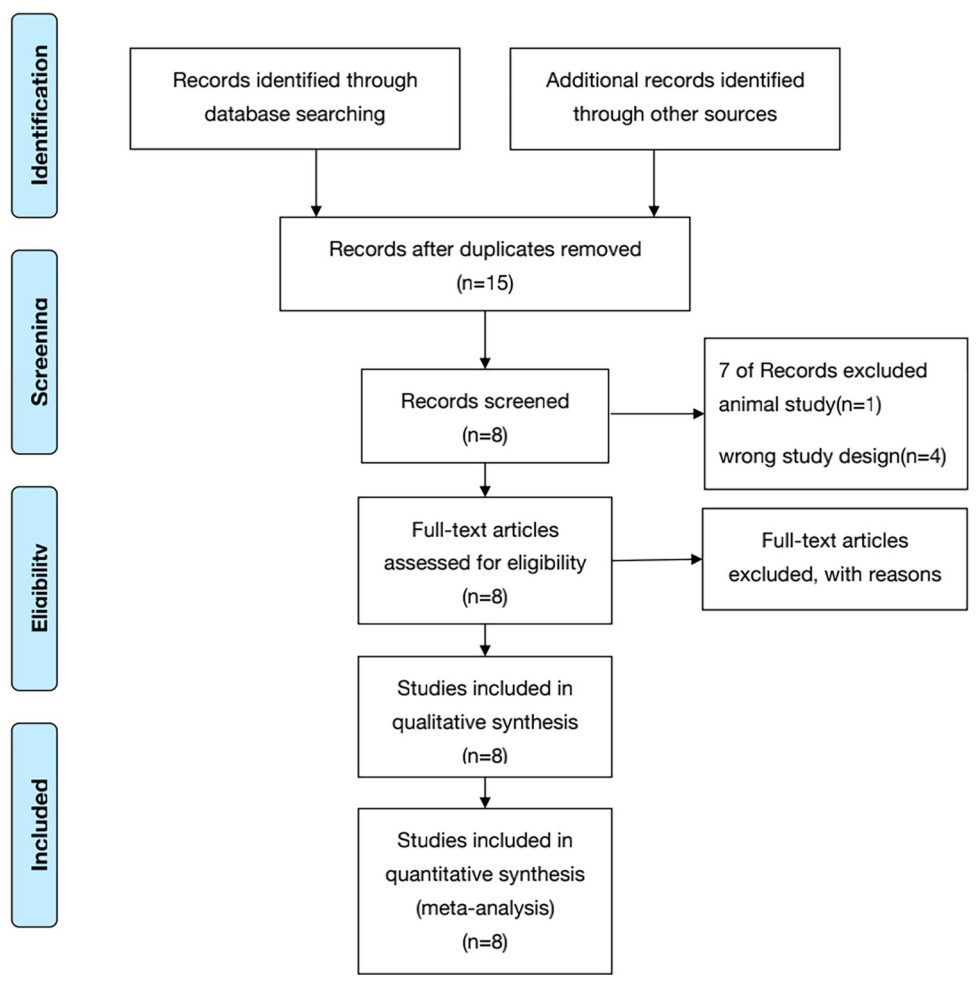

**Fig 1. Flow chart of study selection process.**

of which were administered at 40–60 mg per day with an unknown administration route, while one [25] administered 30 mg orally three times per day. The XBJ dosage was 100 mL in six articles [18–21, 23, 25] and 50 mL in two [22, 24]. XBJ duration was 14 days in seven articles, but not in one study [24]. All studies reported the total effective rate. CRP, IL-6, and IL-10 were reported in five studies [18, 19, 21, 22, 25], and TNF-α in four [18, 19, 21, 22]. None of the eight studies reported adverse events.

## Risk of bias

Five studies [19, 21, 23–25] declared random sequence generation, four [21, 23–25] of which used a random number table, and one [19] the sortition randomization method, so these had low risks. Two trials [18, 20] did not describe random sequence generation, so the risk of bias was regarded as high. Only one trial [22] mentioned a random method but did not describe how random sequences were generated, so the risk of bias was assessed as unclear. All studies had high risks because none described allocation concealment, nor the blinding method. For incomplete outcome data, all studies had a low risk of bias. For selective outcome reporting and other biases, there was no registered protocol or information from the authors, so the risks of bias of all studies were assessed as unclear. Fig 2 shows the risk of bias in all included studies.

**Table 2. Study characteristics.**

| First Author | Publication Year | Country | Sample Size (X/B) | Gender (M/F) | Age (X/B) | Basic Treatment (BT) | Xue-Bi-Jing (XBJ +BT) | Dosage of XBJ | Duration of XBJ (days) | Outcomes | Adverse event report |
|---|---|---|---|---|---|---|---|---|---|---|---|
| Yang et al | 2013 | China | 48 (25/23) | 28/20 | X:51.20 ±10.20 B:50.20 ±9.50 | prednisone (40-60mg/d), antibiotics, oxygen therapy, bronchodilators, nutrition support, normal saline (placebo) | XBJ +BT | 100mL, bid | 14d | Total effective rate, CRP, IL-6, IL-10, TNF-α | No |
| Yang | 2015 | China | 52 (26/26) | 31/21 | X:50.14 ±3.23 B:52.36 ±3.35 | prednisone (40-60mg/d), antibiotics, oxygen therapy, bronchodilators, nutrition support | XBJ +BT | 100mL, bid | 14d | Total effective rate, CRP, IL-6, IL-10, TNF-α | No |
| Zhao | 2016 | China | 140 (70/70) | 68/72 | X:48.10 ±3.54 B:46.90 ±4.05 | prednisone (40-60mg/d), nutrition support, oxygen therapy | XBJ +BT | 100mL, bid | 14d | Total effective rate | No |
| Zhai et al | 2016 | China | 78 (39/39) | 46/32 | X:52.60 ±3.50 B:50.80 ±3.20 | glucocorticoid , nutrition support , antibiotics, oxygen therapy, bronchodilators | XBJ +BT | 50mL, bid | 14d | Total effective rate, CRP, IL-6, IL-10, TNF-α, Temperature, WBC | No |
| Xu | 2016 | China | 68 (34/34) | 39/29 | X:47.10 ±3.80 B:47.20 ±3.60 | prednisone (40-60mg/d), antibiotics, oxygen therapy, bronchodilators, nutrition support | XBJ +BT | 100mL, bid | 14d | Total effective rate, CRP, IL-6, IL-10, TNF-α | No |
| Gu | 2016 | China | 74 (37/37) | 53/21 | 57.30 ±9.40 | prednisone (40-60mg/d), antibiotics, oxygen therapy, bronchodilators, nutrition support | XBJ +BT | 100mL, bid | 14d | Total effective rate, CD4, CD8, CD4/CD8 | No |
| Shi | 2017 | China | 50 (30/20) | 30/20 | X:52.60 ±1.80 B:52.50 ±1.50 | prednisone (30mg po tid), oxygen therapy, antibiotics, nutrition support, normal saline (placebo) | XBJ +BT | 100mL, bid | 14d | Total effective rate, CRP, IL-6, IL-10 | No |
| Kang et al | 2017 | China | 68 (35/33) | 39/29 | X:55.70 ±4.30 B:51.70 ±5.60 | methylprednisolone sodium succinate (80-120mg ivgtt), antibiotics | XBJ +BT | 50mL, bid | - | Total effective rate, WBC, PLT, Neutrophil, HB | No |

Note: Basic Treatment: BT; X: XBJ+BT; B: BT; M: Male; F: Female; bid: two times a day; tid: three times a day; WBC: white blood cell; PLT: platelet; HB: hemoglobin.

## Outcome

**Primary outcome: Total effective rate.** All studies did an objective assessment for RP by the clinical symptoms and the results of lung CT examination according to the clinical efficacy evaluation. The results of the meta-analysis revealed that compared with BT, XBJ+BT significantly increased the total effective rate (n = 578; RR = 1.45, 95% CI: 1.30 to 1.61, p<0.0001, Fig 3A) using the fixed-effects model (heterogeneity: $\chi^2 = 0.95$, $I^2 = 0\%$, p = 1.00).

**Secondary outcomes.** *IL-6*. IL-6 expression was reported in five studies [18, 19, 21, 22, 25]. The results showed that compared to BT, XBJ+BT significantly decreased the expression level of IL-6 (n = 296; MD = -21.56, 95% CI:-27.37 to -15.76, p<0.00001, Fig 3B) using the fixed-effects model (heterogeneity: $\chi^2 = 0.06$, $I^2 = 0\%$, p = 1.00).

*IL-10*. Five studies [18, 19, 21, 22, 25] evaluated IL-10 expression. The results of the fixed-effects (heterogeneity: $\chi^2 = 0.32$, $I^2 = 0\%$, p = 0.99) meta-analysis model showed that XBJ+BT significantly increased IL-10 expression in comparison to BT (n = 296; MD = 17.62, 95% CI:13.95 to 21.29, p<0.00001, Fig 3C).

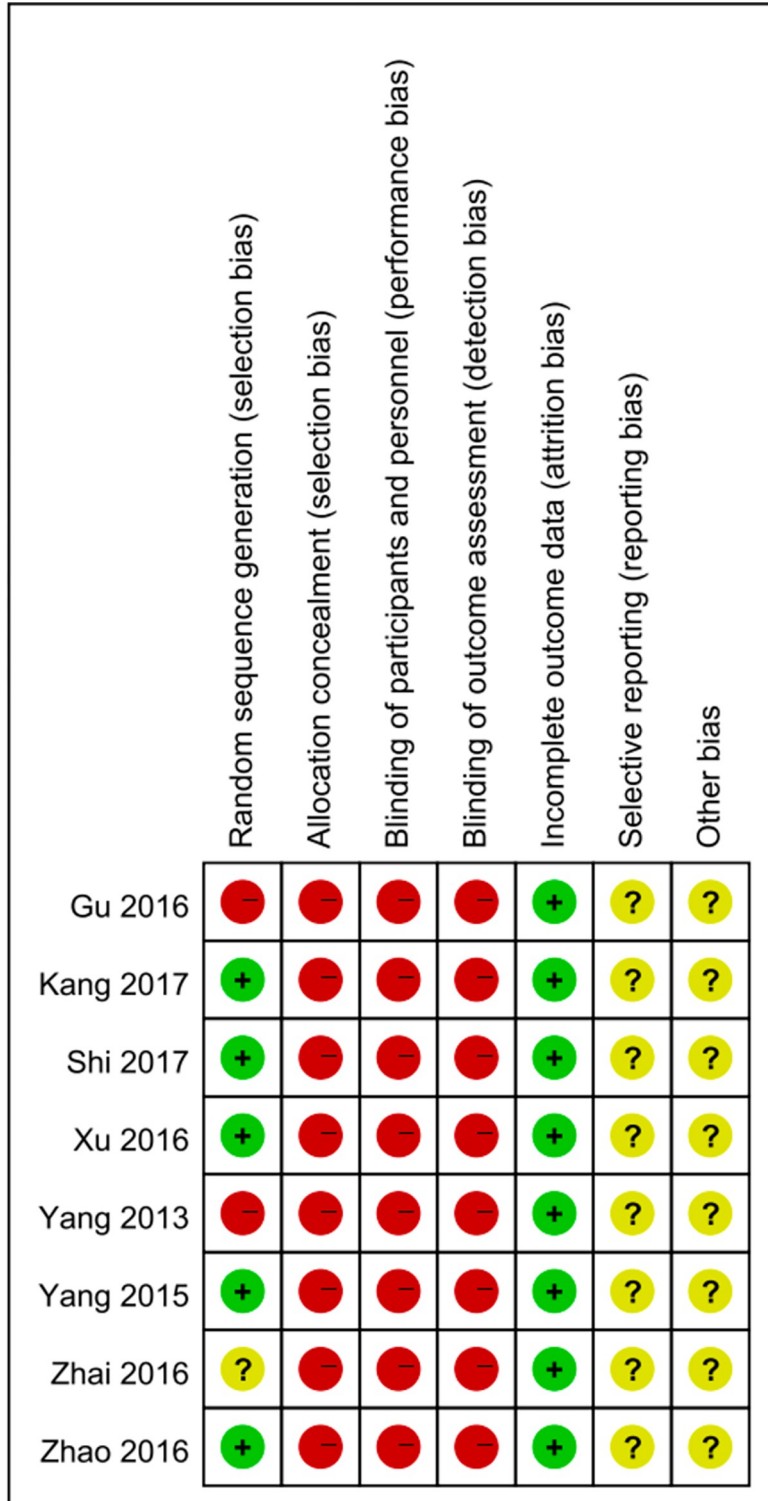

**Fig 2. The risk of bias of all included studies.** The red represents high risk; the green represents low risk; the yellow represents unclear risk.

**A total effective rate**

| Study or Subgroup | XBJ+BT Events | Total | BT Events | Total | Weight | Risk Ratio M-H, Fixed, 95% CI |
|---|---|---|---|---|---|---|
| Gu 2016 | 30 | 37 | 21 | 37 | 12.2% | 1.43 [1.04, 1.97] |
| Kang 2017 | 30 | 35 | 19 | 33 | 11.4% | 1.49 [1.08, 2.06] |
| Shi 2017 | 24 | 25 | 17 | 25 | 9.9% | 1.41 [1.07, 1.87] |
| Xu 2016 | 30 | 34 | 23 | 34 | 13.4% | 1.30 [1.00, 1.70] |
| Yang 2013 | 21 | 25 | 12 | 23 | 7.3% | 1.61 [1.05, 2.47] |
| Yang 2015 | 22 | 26 | 15 | 26 | 8.7% | 1.47 [1.02, 2.12] |
| Zhai 2016 | 32 | 39 | 22 | 39 | 12.8% | 1.45 [1.06, 1.99] |
| Zhao 2016 | 62 | 70 | 42 | 70 | 24.4% | 1.48 [1.20, 1.82] |
| **Total (95% CI)** | | 291 | | 287 | 100.0% | 1.45 [1.30, 1.61] |
| Total events | 251 | | 171 | | | |

Heterogeneity: Chi² = 0.95, df = 7 (P = 1.00); I² = 0%
Test for overall effect: Z = 6.89 (P < 0.00001)

Favours [XBJ+BT]  Favours [BT]

**B IL-6**

| Study or Subgroup | XBJ+BT Mean | SD | Total | BT Mean | SD | Total | Weight | Mean Difference IV, Fixed, 95% CI |
|---|---|---|---|---|---|---|---|---|
| Shi 2017 | 50.5 | 30 | 25 | 70.5 | 25 | 25 | 14.4% | -20.00 [-35.31, -4.69] |
| Xu 2016 | 48.2 | 26.5 | 34 | 70.3 | 20.4 | 34 | 26.7% | -22.10 [-33.34, -10.86] |
| Yang 2013 | 48.35 | 29.64 | 25 | 69.72 | 22.16 | 23 | 15.5% | -21.37 [-36.10, -6.64] |
| Yang 2015 | 48.36 | 29.65 | 26 | 69.73 | 22.17 | 26 | 16.6% | -21.37 [-35.60, -7.14] |
| Zhai 2016 | 48.52 | 28.46 | 39 | 70.62 | 21.64 | 39 | 26.8% | -22.10 [-33.32, -10.88] |
| **Total (95% CI)** | | | 149 | | | 147 | 100.0% | -21.56 [-27.37, -15.76] |

Heterogeneity: Chi² = 0.06, df = 4 (P = 1.00); I² = 0%
Test for overall effect: Z = 7.28 (P < 0.00001)

Favours [XBJ+BT]  Favours [BT]

**C IL-10**

| Study or Subgroup | XBJ+BT Mean | SD | Total | BT Mean | SD | Total | Weight | Mean Difference IV, Fixed, 95% CI |
|---|---|---|---|---|---|---|---|---|
| Shi 2017 | 80.5 | 15 | 25 | 60.5 | 18 | 25 | 16.0% | 20.00 [10.82, 29.18] |
| Xu 2016 | 79.8 | 15.1 | 34 | 62.3 | 16.4 | 34 | 24.0% | 17.50 [10.01, 24.99] |
| Yang 2013 | 79.56 | 14.59 | 25 | 62.46 | 17.37 | 23 | 16.2% | 17.10 [7.98, 26.22] |
| Yang 2015 | 79.57 | 14.6 | 26 | 62.47 | 17.38 | 26 | 17.7% | 17.10 [8.38, 25.82] |
| Zhai 2016 | 80.2 | 14.91 | 39 | 63.25 | 17.41 | 39 | 26.1% | 16.95 [9.76, 24.14] |
| **Total (95% CI)** | | | 149 | | | 147 | 100.0% | 17.62 [13.95, 21.29] |

Heterogeneity: Chi² = 0.32, df = 4 (P = 0.99); I² = 0%
Test for overall effect: Z = 9.40 (P < 0.00001)

Favours [XBJ+BT]  Favours [BT]

**D TNF-α**

| Study or Subgroup | XBJ+BT Mean | SD | Total | BT Mean | SD | Total | Weight | Mean Difference IV, Fixed, 95% CI |
|---|---|---|---|---|---|---|---|---|
| Xu 2016 | 51.8 | 18.6 | 34 | 77.2 | 21.4 | 34 | 29.0% | -25.40 [-34.93, -15.87] |
| Yang 2013 | 52.26 | 19.57 | 25 | 77.12 | 22.74 | 23 | 18.2% | -24.86 [-36.91, -12.81] |
| Yang 2015 | 52.27 | 19.58 | 26 | 77.13 | 22.75 | 26 | 19.8% | -24.86 [-36.40, -13.32] |
| Zhai 2016 | 51.63 | 18.76 | 39 | 78.36 | 21.45 | 39 | 33.0% | -26.73 [-35.67, -17.79] |
| **Total (95% CI)** | | | 124 | | | 122 | 100.0% | -25.63 [-30.77, -20.50] |

Heterogeneity: Chi² = 0.09, df = 3 (P = 0.99); I² = 0%
Test for overall effect: Z = 9.78 (P < 0.00001)

Favours [XBJ+BT]  Favours [BT]

**E CRP**

| Study or Subgroup | XBJ+BT Mean | SD | Total | BT Mean | SD | Total | Weight | Mean Difference IV, Random, 95% CI |
|---|---|---|---|---|---|---|---|---|
| Shi 2017 | 75.5 | 30 | 25 | 125.5 | 38 | 25 | 17.2% | -50.00 [-68.98, -31.02] |
| Xu 2016 | 73.7 | 27.5 | 34 | 122.4 | 38.8 | 34 | 24.3% | -48.70 [-64.69, -32.71] |
| Yang 2013 | 73.45 | 29.46 | 25 | 121.84 | 40.99 | 23 | 15.0% | -48.39 [-68.74, -28.04] |
| Yang 2015 | 73.46 | 29.47 | 26 | 121.85 | 40.98 | 26 | 16.5% | -48.39 [-67.79, -28.99] |
| Zhai 2016 | 73.53 | 29.21 | 39 | 121.43 | 38.42 | 39 | 27.0% | -47.90 [-63.05, -32.75] |
| **Total (95% CI)** | | | 149 | | | 147 | 100.0% | -48.61 [-56.49, -40.73] |

Heterogeneity: Tau² = 0.00; Chi² = 0.03, df = 4 (P = 1.00); I² = 0%
Test for overall effect: Z = 12.10 (P < 0.00001)

Favours [XBJ+BT]  Favours [BT]

**Fig 3. A.** Forest plot of total effective rate of meta-analysis of XBJ+BT group VS BT group. CI: confidence interval. Risk Ratio was the effect size. We made Mantel–Haenszel (M-H) fixed-effects. **B.** Forest plot of IL-6 expression of meta-analysis of XBJ +BT group VS BT group. CI: confidence interval. Mean Difference was the effect size. We made inverse variance (IV) fixed-effects. **C.** Forest plot of IL-10 expression of meta-analysis of XBJ+BT group VS BT group. CI: confidence interval. Mean Difference was the effect size. We made inverse variance (IV) fixed-effects. **D.** Forest plot of TNF-α expression of meta-analysis of XBJ+BT group VS BT group. CI: confidence interval. Mean Difference was the effect size. We made inverse variance (IV) fixed-effects. **E.** Forest plot of CRP expression of meta-analysis of XBJ+BT group VS BT group. CI: confidence interval. Mean Difference was the effect size. We made inverse variance (IV) fixed-effects.

*TNF-α.* Four trials [18, 19, 21, 22] analyzed TNF-α levels. In contrast to BT, the fixed-effects model (heterogeneity: $\chi^2 = 0.09$, $I^2 = 0\%$, $p = 0.99$) showed a significant decrease of TNF-α expression in XBJ+BT (n = 246; MD = -25.63, 95% CI:-30.77 to -20.50, $p<0.00001$, Fig 3D).

*CRP.* Five trials [18, 19, 21, 22, 25] assessed CRP expression. Compared with BT, the level of CRP was significantly decreased in XBJ+BJ (n = 296; MD = − -48.61, 95% CI: − -56.49, − -40.73, $p<0.00001$, Fig 3E), without significant heterogeneity ($\chi^2 = 0.03$, $I^2 = 0\%$, $p = 1.00$).

*AEs.* As none of these trials reported AEs, we could not perform this meta-analysis.

*Publication bias.* Because there were less than10 articles for meta-analysis, a funnel plot to evaluate publication bias was not generated. In order to evaluate publication bias more accurately, egger test, a quantitative analysis method was performed by stata software (Version 15). The results showed that p-value was equal to 0.845, indicating that the risk of publication bias was low.

## Discussion

XBJ is a TCM injection that was approved by the State Food and Drug Administration (China FDA) in 2004 and is mainly used in systemic inflammatory response syndrome and sepsis in China. XBJ is a combination of Carthamus tinctorius flowers (Honghua in Chinese), Paeonia lactiflora roots (Chishao), Ligusticum chuanxiong rhizomes (Chuanxiong), Angelica sinensis roots (Danggui), and Salvia miltiorrhiza roots (Danshen). The main components of XBJ detected by ultra-high performance liquid chromatography include Hydroxysafflor yellow A, Oxypaeoniflorin, Senkyunolide I and Benzoylpaeoniflorin [26]. Studies have shown that XBJ mainly inhibits the activation of inflammatory pathways by inhibiting TNF-α, IL-1, IL-6, IL-8, IL-17 and PAR-1 and improves coagulation dysfunction in sepsis [27–29]. There is evidence that XBJ has high safety, mild adverse reactions and good outcomes when used reasonably in clinical practice [30].

To the best of our knowledge, this is the first meta-analysis of RCTs on XBJ for the treatment of RP. In this study, eight RCTs were screened and included in the quantitative synthesis, with a total of 578 participants, including 296 in the XBJ+BT group and 282 in the BT group. Based on current clinical evidence, the results of this meta-analysis indicate that XBJ+BT significantly increased the total effective rate, IL-10, and decreased IL-6, TNF-α, and CRP levels compared to BT. Since no AEs were reported in any trials, the safety of XBJ for RP treatment remains unclear. Furthermore, according to these RCTs, the recommended XBJ dosage was 100 mL for 14 days, and the prednisone dosage 40–60 mg per day for 14 days.

TNF-α is a cytokine that initiates inflammatory responses. Research [31] suggests that TNF-α is involved in the formation of RP and early reduction of TNF-α level can alleviate RP. IL-6 is a inflammatory factor secreted by Th2 cells. Current study suggests that the expression of IL-6 in the serum of patients with RP is significantly increased, which can be used as a predictive indicator for RP [32]. It has been reported that concentration of CRP is a very sensitive and effective indicator for judging the occurrence, development and prognosis of patients with RP [33]. IL-10 is known as cytokine synthesis inhibitor. IL-10 can inhibit the expression of

inflammatory factors such as TNF, IL-6, and IL-1 through activated macrophages. As the most important anti-inflammatory cytokine in the body, its early expression in RP acts as a negative feedback to inhibit the occurrence of RP [34]. Our results showed that XBJ+BT significantly decreased pro-inflammatory factors such as IL-6, TNF-α and CRP levels and increased anti-inflammatory cytokine IL-10 to alleviate RP compared to BT.

For heterogeneity, we conducted forest plot from which we may find that the heterogeneity was low as $I^2$ value < 50%. Nonetheless, for the degree of heterogeneity of each study, sensitivity analysis as well as changing the model was also implemented to evaluate in depth. First of all, each study was removed one by one to estimate whether the results could have been affected. The results showed that $I^2$ was less than 50% for any one study, suggesting that the heterogeneity of each study was low. Then, the fixed-effects model was changed to the random-effects model and the p-value was consistent and without apparent fluctuation, suggesting that the heterogeneity of each study was low. So we believed that this analysis confirmed the stability of our results.

## Limitations

There were some limitations to this study. It is well known that in CAM clinical trials, poor methodological quality exists [35]. All trials had a high risk of bias. None of the included studies described how allocation concealment was performed, whether blinding was implemented, and what kind of blinding method was employed. In terms of reporting bias, whether there was reporting bias was not clear because no protocol or information could be found for reference. As a result, all trials were of low quality. Even after conducting the meta-analysis, the results were not convincing.

Despite a comprehensive search of seven English and Chinese databases, including PubMed and CNKI, as well as clinical trial registries, there may be studies meeting the inclusion criteria that were not included. In addition, the eight included trials were all written in Chinese, so it was difficult to avoid the possibility of selective bias.

Moreover, the occurrence of AEs during the TCM treatment process should be paid close attention to. AEs caused by TCM injections have also been reported. In this study, we considered the occurrence of AEs as a secondary outcome, but unfortunately, none of the eight included trials clearly showed whether AEs occurred or not.

## Conclusion

Based on the present results, we do not recommend XBJ as an adjuvant treatment for RP. However, we think that XBJ has the potential as an adjuvant treatment for RP. Therefore, future randomized controlled trials with rigorous design, strict implementation, and standard reporting are be needed to further evaluate the efficacy and safety of XBJ in the adjuvant treatment of RP.

## Supporting information

**S1 Checklist. PRISMA 2009 checklist.**
(DOC)

**S1 File. INPLASY protocol.** The Efficacy and Safety of Xuebijing Injection in the Treatment of Radiation Pneumonitis: A Protocol for Systematic Review and Meta-Analysis.
(PDF)

## Author Contributions

**Conceptualization:** Ying Zhang, Xueqian Wang, Wei Hou.

**Methodology:** Shuo Wang, Yuxiao Li.

**Project administration:** Yuerong Gui, Jun Dong.

**Writing – original draft:** Zheng Li, Dandan Wang.

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
