## [Decision Letter · Decision Letter 0]

24 Feb 2022

PONE-D-21-23587Efficacy and Safety of Xuebijing Injection for Radiation Pneumonitis: A Meta-AnalysisPLOS ONE

Dear Dr. Li,

Thank you for submitting your manuscript to PLOS ONE. After careful consideration, we feel that it has merit but does not fully meet PLOS ONE’s publication criteria as it currently stands. Therefore, we invite you to submit a revised version of the manuscript that addresses the points raised during the review process.

The mansucript has been seen by three reviewers and their comments may be seen below.

Overall the reviewers believe that the meta analysis is well conducted, however have expressed concerns regarding the high risk of bias in the clinical trials included in the analysis, therefore we believe that the discussion section should be further elaborated to discuss this issue. Furthermore we recommend that Forest Plot legends are annotated to provide additional details for the readers.

Could you please revise the manuscript to carefully address the concerns raised?

We look forward to receiving your revised manuscript.

Kind regards,

Lucinda Shen, MSc

Staff Editor

PLOS ONE

https://journals.plos.org/plosone/s/fileid=ba62/PLOSOne_formatting_sample_title_authors_affiliations.pdf".

Reviewers' comments:

Reviewer's Responses to Questions

**Comments to the Author**

1. Is the manuscript technically sound, and do the data support the conclusions?

Reviewer #1: Yes

Reviewer #2: No

Reviewer #3: Yes

2. Has the statistical analysis been performed appropriately and rigorously? 

Reviewer #1: Yes

Reviewer #2: No

Reviewer #3: Yes

3. Have the authors made all data underlying the findings in their manuscript fully available?

Reviewer #1: Yes

Reviewer #2: Yes

Reviewer #3: Yes

4. Is the manuscript presented in an intelligible fashion and written in standard English?

Reviewer #1: Yes

Reviewer #2: Yes

Reviewer #3: Yes

5. Review Comments to the Author

Reviewer #1: thorough review of the literature. data and analysis sound.

appreciate the authors despite the favorable data to realize that the limitations of the current literature.

conclusion drawn from meta analysis is sound

Reviewer #2: This manuscript is easily readable, concise, and well-written. However, as you stated in Limitations section, all trials which included in this meta-analysis had a high risk of bias. None of the included studies described how allocation concealment was performed, whether blinding was implemented, and what kind of blinding method was employed. In terms of reporting bias, whether there was reporting bias was not clear because no protocol or information could be found for references. As a result, all trials were of low quality. Therefore, we cannot acquire any meaningful conclusions from these trials. You should conduct the meta-analysis with high quality trials. If there is no high quality trials in certain clinical field, there is no point to carry out the meta-analysis. Conclusions of this study has no credibility.

Reviewer #3: The manuscript submitted by the authors describe about the metaanalysis of 8 studies describing an agent Xuibijing as a radiation protector against radiation pneumonitis.

The authors have only taken randomised clinical trial for the purpose of metananalysis and have omitted studies which were preclinical in nature and studies which were retrospective.

The authors have followed the PRISMA guidelines for the metaanalysis and have properly done the risk bias assessment of all the studies as represented by the traffic light plot.

The authors have used appropriate statististical tests for analysing the data from the RCT’s.

Although the forest plots show a favourable response with the use of the experimental drug in terms of various diagnostic metrics, even then the authors do not recommend the use of the agent in question for radiation pneumonits.

The manuscript is well written and adheres strictly to the guidelines for reporting meta-analysis. Even though the metaanalysis can be termed as negative against the use of the experimental agent, I would still recommend the publication of the study after major modifications.

Comments

1. The authors in the discussion section need to elaborate a bit more on this agent as its mechanism of action, its pharmacology, kinetics and its toxicities have not been extensively reported given that it is a Chinese medicine and the readers outside China will not have much knowledge about it.

2. What is a bit surprising to me is that the clinical end points and majority of the biological markers (inflammatory cytokines) showed an improvement on the use of combination of the experimental drug + conventional therapy as opposed to the conventional therapy alone except for IL10 which is a proinflammatory cytokine which showed an increased levels which is a bit counterintuitive? An explanation needs to b e give regarding this phenomenon in the discussion section.

3. In the methodology section: the authors should write in detail if any of the studies did an objective assessment for radiation pneumonitis.

4. The authors have shown in their forest plots the results of the RCT’s favouring the use of xuibijing against RP as none of the plots were straddling the unity line. However, based on the limitation s of the RCT’s, the authors in their concluding remarks have not recommended the use of this agent against RP. I would expect the authors to clearly annotate the forest plots and expand the legends of figure 3. I would also want the authors to clearly mention the diagnostic metrics they used in the forest plots.(was it the Hazard ratio or the odds ratio?)

5. The authors also mention that none of the studies mentioned anything about AE’s & SAE’s in the limitation of the studies. In this last paragraph, the authors mention about TCM injection. The authors write in detail what TCM stands for as it has not been mentioned previously in the manuscript.

6. The authors mentioned that they did not do a funnel plot as the number of studies were less than 10. However, it will be worthwhile to do the HSROC curves to depict the extent of heterogeinity of each study.

6. PLOS authors have the option to publish the peer review history of their article (what does this mean?). If published, this will include your full peer review and any attached files.

Reviewer #1: No

Reviewer #2: No

Reviewer #3: No

---

## [Author Response · Author response to Decision Letter 0]

30 Apr 2022

Dear Editors and Reviewers,

Thanks very much for taking your time to review this manuscript. We really appreciate all your comments and suggestions!These suggestions have enabled us to improve our work. The comments are reproduced and our responses are given directly afterward in a different color (in blue). We would like also to thank you for allowing us to resubmit a revised copy of the manuscript.

Suggestions from reviewers:

Reviewer #1: thorough review of the literature. data and analysis sound.appreciate the authors despite the favorable data to realize that the limitations of the current literature.conclusion drawn from meta analysis is sound

Reply to reviewer #1: We appreciate your careful review and approval of our manuscript.

Reviewer #2: This manuscript is easily readable, concise, and well-written. However, as you stated in Limitations section, all trials which included in this meta-analysis had a high risk of bias. None of the included studies described how allocation concealment was performed, whether blinding was implemented, and what kind of blinding method was employed. In terms of reporting bias, whether there was reporting bias was not clear because no protocol or information could be found for references. As a result, all trials were of low quality. Therefore, we cannot acquire any meaningful conclusions from these trials. You should conduct the meta-analysis with high quality trials. If there is no high quality trials in certain clinical field, there is no point to carry out the meta-analysis. Conclusions of this study has no credibility.

Reply to reviewer #2: Thank you for your support of our manuscript. We agree with you that all of the trials you mentioned have a high risk of bias. We followed the PRISMA guidelines for the meta-analysis and have rigorously performed the risk bias assessment of all the trial. It is for this reason that we do not recommend XBJ as a treatment for RP, despite the good results shown by the forest plot. We are also concerned that the conclusions drawn are inaccurate due to the high risk of bias. But we think this article is still relevant.The first is that based on the rigorous process of evaluation, we do not recommend XBJ for RP treatment. Second, due to the lack of more effective drugs with less side effects to treat RP, XBJ has still some potential in the treatment of RP. Third, this paper provides some ideas and references for more rigorous clinical trials in the future, so as to obtain high-quality clinical evidence.

Reviewer #3: The manuscript submitted by the authors describe about the metaanalysis of 8 studies describing an agent Xuibijing as a radiation protector against radiation pneumonitis.

The authors have only taken randomised clinical trial for the purpose of metananalysis and have omitted studies which were preclinical in nature and studies which were retrospective.

The authors have followed the PRISMA guidelines for the metaanalysis and have properly done the risk bias assessment of all the studies as represented by the traffic light plot.

The authors have used appropriate statististical tests for analysing the data from the RCT’s.

Although the forest plots show a favourable response with the use of the experimental drug in terms of various diagnostic metrics, even then the authors do not recommend the use of the agent in question for radiation pneumonits.

The manuscript is well written and adheres strictly to the guidelines for reporting meta-analysis. Even though the metaanalysis can be termed as negative against the use of the experimental agent, I would still recommend the publication of the study after major modifications.

Reply to reviewer #3: Thank you for your patience in reviewing our manuscript and itemizing key points in our manuscript. We will revise according to the comments in order to improve our manuscript and meet the requirements of the journal.

Comments to the author

1. The authors in the discussion section need to elaborate a bit more on this agent as its mechanism of action, its pharmacology, kinetics and its toxicities have not been extensively reported given that it is a Chinese medicine and the readers outside China will not have much knowledge about it.

Answer: Sure. We added more information about XBJ in the discussion part to let researchers and readers better know XBJ. 

2. What is a bit surprising to me is that the clinical end points and majority of the biological markers (inflammatory cytokines) showed an improvement on the use of combination of the experimental drug + conventional therapy as opposed to the conventional therapy alone except for IL10 which is a proinflammatory cytokine which showed an increased levels which is a bit counterintuitive? An explanation needs to b e give regarding this phenomenon in the discussion section.

Answer: Thank you for pointing this phenomenon out. Let us explain this phenomenon. IL-10 is known as cytokine synthesis inhibitor. IL-10 can inhibit the expression of inflammatory factors such as TNF, IL-6, and IL-1 through activated macrophages. As the most important anti-inflammatory cytokine in the body, its early expression in RP acts as a negative feedback to inhibit the occurrence of RP. So, IL-10 is a kind of anti-inflammatory cytokines which is different from other pro-inflammatory factors such as IL-6, TNF and IL-1, etc. XBJ+BT significantly decreased pro-inflammatory factors such as IL-6, TNF-α and CRP levels and increased anti-inflammatory cytokine IL-10 to alleviate RP compared to BT. 

3. In the methodology section: the authors should write in detail if any of the studies did an objective assessment for radiation pneumonitis.

Answer: Thank you for mentioning this key point. We added the clinical efficacy evaluation in ‘the type of outcome’ part. We also mentioned that all studies did an objective assessment for RP by the clinical symptoms and the results of lung CT examination according to the clinical efficacy evaluation in the ‘outcome’ part.

4. The authors have shown in their forest plots the results of the RCT’s favouring the use of xuibijing against RP as none of the plots were straddling the unity line. However, based on the limitation s of the RCT’s, the authors in their concluding remarks have not recommended the use of this agent against RP. I would expect the authors to clearly annotate the forest plots and expand the legends of figure 3. I would also want the authors to clearly mention the diagnostic metrics they used in the forest plots.(was it the Hazard ratio or the odds ratio?)

Answer: Thank you for your attention to the details of the forest plot. We apologized for not annotating the forest plot clearly. We re-annotated the forest plot and expanded the legend of Figure 3. For the total effective rate，risk ratio (RR) was used as effect size. For cytokines or inflammatory factors，mean difference (MD) was the effect size. 

5. The authors also mention that none of the studies mentioned anything about AE’s & SAE’s in the limitation of the studies. In this last paragraph, the authors mention about TCM injection. The authors write in detail what TCM stands for as it has not been mentioned previously in the manuscript.

Answer: Sure. We explained what was TCM in detail so that readers may clearly understand the meaning of TCM.

6. The authors mentioned that they did not do a funnel plot as the number of studies were less than However, it will be worthwhile to do the HSROC curves to depict the extent of heterogeinity of each study.

Answer: Thank you for raising this method to address the heterogeneity. We have been thinking about this issue for a long time and have also consulted related paper. We suspect that the editors need us to discuss the heterogeneity of each study. The HSROC curve seems to be used to evaluate the sensitivity and specificity of diagnostic indicators. We have consulted a large number of literature, but unfortunately, we couldn't figure out how to use the HSROC curve to describe the heterogeneity of each study. So we tried to take the following two approaches instead if it is feasible. For heterogeneity, we conducted forest plot from which we may find that the heterogeneity was low as I² value < 50 %. Nonetheless, for the degree of heterogeneity of each study, sensitivity analysis as well as changing the model was also implemented to evaluate in depth. With both methods, the heterogeneity of the studies was low. So we believed that this analysis confirmed the stability of our results.What's more, in order to evaluate publication bias more accurately, egger test, a quantitative analysis method was performed by stata software (Version 15). The results showed that p-value was equal to 0.845, indicating that the risk of publication bias was low. 

If our understanding of this issue deviates from your ideas, we hope that the editors and reviewers will give us another chance to revise this to improve the manuscript. Thanks again.

---

## [Editor Report · Decision Letter 1]

10 May 2022

Efficacy and Safety of Xuebijing Injection for Radiation Pneumonitis: A Meta-Analysis

PONE-D-21-23587R1

Dear Dr. Li,

We’re pleased to inform you that your manuscript has been judged scientifically suitable for publication and will be formally accepted for publication once it meets all outstanding technical requirements.

Kind regards,

Jayant s Goda

Guest Editor

PLOS ONE
---

## [Editor Report · Acceptance letter]

23 May 2022

PONE-D-21-23587R1 

Efficacy and Safety of Xuebijing Injection for Radiation Pneumonitis: A Meta-Analysis 

Dear Dr. Hou:

I'm pleased to inform you that your manuscript has been deemed suitable for publication in PLOS ONE. Congratulations! Your manuscript is now with our production department. 

Kind regards, 

on behalf of

Dr. Jayant s Goda 

Guest Editor

PLOS ONE